# The Power of Play: Strategies for Enhancing Development in Children with Autism Spectrum Disorders

**DOI:** 10.3390/s24206720

**Published:** 2024-10-19

**Authors:** Felix-Constantin Adochiei, Simona-Narcisa Arghir, Ioana Raluca Adochiei, Florin Ciprian Argatu, George Calin Seritan, Bogdan Alexandrescu

**Affiliations:** 1Faculty of Electrical Engineering, National University of Science and Technology Politehnica Bucharest, 313 Splaiul Independentei, 060042 Bucharest, Romania; felix.adochiei@upb.ro (F.-C.A.); simona.arghir@stud.fim.upb.ro (S.-N.A.); florin.argatu@upb.ro (F.C.A.); george.seritan@upb.ro (G.C.S.); 2Academy of Romanian Scientists, Ilfov 3, 050044 Bucharest, Romania; 3Emil Palade Center of Excellence for Young Researchers, Academy of Romanian Scientists, Ilfov 3, 050044 Bucharest, Romania; 4Faculty of Aircraft and Military Vehicles, Military Technical Academy, Ferdinand I”, 050141 Bucharest, Romania; 5Faculty of Electronics, Telecommunications and Information Technology, National University of Science and Technology Politehnica Bucharest, 313 Splaiul Independentei, 060042 Bucharest, Romania; bogdan.alexandrescu@upb.ro

**Keywords:** autism, Makey Makey, Scratch, therapeutic strategy, emotional development

## Abstract

The increasing prevalence of autism spectrum disorder (ASD) underscores its significant impact on individuals and the importance of early intervention. ASD affects various aspects of life, including education, emotional development, and social interactions. Besides traditional therapeutic approaches, a novel strategy incorporating digital games has been introduced. Four games developed using Makey Makey and Scratch aim to enhance cognitive skills in children with ASD. This approach facilitates emotional and intellectual development, tracks progress, and offers personalized and engaging interventions. This study demonstrated significant improvements in memory and concentration among participants, with an average improvement of 23.38 points. The most notable enhancements were observed in children aged 10, who showed an average improvement of 25.67 points. Additionally, female participants exhibited a slightly higher average improvement compared to males. The Memory Maze game also effectively enhanced cognitive skills in children with different types of ADHD. Children with the Hyperactive–Impulsive type showed the highest average improvement, with 25.00 points, followed by those with the Combined type, with 24.15 points, and the Inattentive type, with 23.53 points. These findings highlight the potential use of these tools in both ASD and ADHD therapy, providing a structured and enjoyable learning environment that supports cognitive development and therapeutic outcomes.

## 1. Introduction

Autism spectrum disorder (ASD) is a prevalent neurodevelopmental condition characterized by deficits in social communication and interaction, along with restricted and repetitive patterns of behavior, interests, or activities [1]. The global prevalence of ASD is estimated to be approximately 1 in 100 children, though this figure varies due to differences in diagnostic practices and awareness across regions [2]. 

A significant study from 2016 highlighted the notably high identification rate of autism in Romania, with 14.3% of children having been diagnosed [3]. This finding has played a crucial role in raising awareness, influencing national health policies, and driving improvements in education and support services for children with ASD.

This underscores the necessity for increased awareness and adaptation of health, educational, and social resources to better meet the needs of individuals with ASD. By tailoring resources to accommodate the growing number of diagnosed cases, policymakers and educators can offer more effective support systems and interventions for affected children and their families.

The etiology of ASD involves complex interactions between genetic and environmental factors. Recent research highlights that prenatal stress, birth complications, and maternal health conditions significantly contribute to abnormal neurological development, potentially leading to ASD [4]. Genetic studies have identified numerous susceptibility genes, revealing a heritable component to the disorder [5]. Moreover, a study conducted by Yale University in 2023 identified two distinct developmental pathways to autism, suggesting that gene mutations associated with the disorder can lead to different neurodevelopmental outcomes [6]. This emphasizes the complexity of ASD and the need for a nuanced understanding of its genetic underpinnings.

The symptoms of autism are diverse, ranging from delayed speech and limited eye contact to repetitive behaviors and difficulties in social interactions. Clinical diagnosis typically occurs in early childhood, often before age three, based on developmental assessments and behavioral observations [7]. Early intervention is critical, as it can significantly influence the developmental trajectory and improve long-term outcomes for children with ASD [8]. Evidence-based interventions, such as Applied Behavior Analysis (ABA), speech therapy, and occupational therapy, have enhanced communication skills, social skills, and adaptive behaviors [9]. These interventions underscore the necessity of early, intensive, and individualized therapeutic strategies to address the unique challenges faced by individuals with ASD.

This study aims to evaluate the effectiveness of the Memory Maze game in improving memory and concentration among children with ASD. This research uses a randomized control trial with 60 participants and explores how ADHD comorbidities may influence the results of the intervention.

## 2. The State of the Art

Individuals with autism spectrum disorder (ASD) commonly experience challenges in four primary domains: social skills, emotional regulation, attention control, and cognitive abilities (Figure 1) [1]. These difficulties can significantly impact their ability to function effectively in social settings, maintain concentration, and engage in learning activities over extended periods.

Current therapeutic approaches aim to alleviate symptoms and improve quality of life for individuals with ASD rather than seek a cure. Personalized therapy plans, tailored to the specific needs of each child, are essential given the wide variability in ASD manifestations. Applied Behavior Analysis (ABA) is a cornerstone of behavioral intervention, utilizing reinforcement strategies to enhance socially significant behaviors through systematic and individualized methods [6]. Other effective interventions include speech therapy, which addresses communication deficits; cognitive behavioral therapy (CBT), which helps manage anxiety and improve emotional regulation; and social skills training, which enhances interaction capabilities [7]. Occupational therapy, developmental play therapy, and floortime therapy also support the development of sensory processing, motor skills, and emotional connections [8].

Recent advancements have integrated digital and electronic applications into therapeutic practices, providing innovative solutions for managing ADHD and ASD symptoms. For example, Akili Interactive’s EndeavorRx, the first video game approved by the FDA for ADHD treatment, has led to improvements in attention and impulse control, suggesting potential applications for ASD [9]. Cogmed Working Memory Training and Play Attention utilize cognitive training and neurofeedback techniques to enhance attention and working memory in children with ADHD, indicating a promising overlap in therapeutic strategies for ASD and ADHD [10,11]. Neurofeedback Therapy and BrainTrain software (2020.2) offer personalized cognitive training aimed at improving executive functions, while MediTrain, a digital meditation application, enhances attention and impulse control [12,13,14]. Comprehensive platforms such as ADHD 365, Lumosity, MyADHD, and GoNoodle integrate educational, cognitive, and mindfulness activities, supporting holistic development and emotional regulation [15,16,17,18].

Moreover, recent studies have shown that virtual reality (VR) and augmented reality (AR) applications are practical in enhancing social skills and emotional regulation in children with ASD. For instance, a study by Dechsling et al. (2021) demonstrated that VR-based naturalistic developmental behavioral interventions significantly improved joint attention and social interactions in children with ASD [19]. Similarly, Ip et al. (2018) found that a VR-enabled approach enhanced emotional and social adaptation skills in children with ASD [20]. These findings suggest that integrating advanced technological tools with traditional therapeutic methods can provide a more comprehensive approach to ASD treatment.

Differentiating between types of autism is also crucial for the tailoring of effective interventions. Research has identified several subtypes of autism, each with unique behavioral and developmental profiles. Recognizing these distinctions allows for more precise therapeutic strategies, addressing specific needs and optimizing outcomes for each individual. This differentiation underscores the complexity of ASD and the importance of personalized, evidence-based treatment approaches.

In addition to traditional approaches, we propose a novel therapeutic strategy incorporating digital games: an integrated platform with four games developed using Makey Makey and Scratch to enhance cognitive skills in children with ASD. This approach facilitates emotional and intellectual development, tracks progress, and offers personalized, engaging interventions. Preliminary results have demonstrated significant improvements in memory and concentration skills among participants, highlighting the potential of these tools in ASD therapy.

The integration of these technological advancements with traditional therapeutic approaches provides a multifaceted strategy to support individuals with ASD, facilitating their participation in daily activities and improving their overall quality of life. Establishing personalized learning objectives and employing effective teaching strategies remain central to these interventions, which highlight the importance of individualized, evidence-based approaches in ASD treatment.

In line with the framework proposed by Moreno et al. (2023) for accessible software design [21], we are actively adapting the Memory Maze game to ensure it meets the specific needs of children with autism. Our focus on user adaptability and accessibility is reflected in the game’s ability to adjust to different cognitive abilities and autism profiles. These principles of accessibility are integral to the game’s design, ensuring that it is a practical and effective tool for children with ASD.

Also, in line with the principles outlined by Moreno et al. (2022) on Human-Centered Design (HCD) [22] for users with autism, we have begun integrating personalization features into the Memory Maze game, targeting specific types of autism and cognitive challenges. While the current version of the game has been developed based on input from therapists and educators, future iterations will more formally incorporate HCD techniques such as user research, prototyping, and iterative testing to better align with the individual needs and capabilities of children with different types of ASD. We acknowledge the importance of continuously improving the game design process and will seek to further apply HCD principles in future developments.

## 3. Materials and Methods

To address the needs of autistic children, a sophisticated system leveraging the v1.6 Makey Makey development kit, manufactured by JoyLabz, located in Santa Cruz, CA, USA, and 3.0 Scratch programming language, developed by the Massachusetts Institute of Technology (MIT) Media Lab, located in Cambridge, MA, USA, was devised to enhance attention and fine motor skills. This innovative system utilizes electronic components that connect to various everyday or specialized objects, transforming them into touch-sensitive input devices. The system’s core is the Makey Makey kit (Figure 2), which operates on Arduino Leonardo Rev3 firmware and communicates with computers via the Human Interface Device (HID) protocol, ensuring seamless integration across various applications [23].

The system’s architecture allows conductive materials—such as fruits, play dough, or pencil drawings—to be linked to dedicated pins and connection cables equipped with alligator clips. When these materials are touched, the circuit completes, and the microcontroller interprets the input as a keyboard key press or mouse click. This setup promotes hands-on learning and the development of fine motor skills, offering a versatile platform for customized educational experiences. The critical components of the Makey Makey system include the main circuit board, a USB connector for interfacing with a computer, and connection cables that integrate various conductive objects, creating a personalized and immersive learning environment for children with autism.

Scratch was selected for developing interactive and engaging educational tools for autistic children due to its user-friendly visual programming language, developed by the Massachusetts Institute of Technology (MIT), located in Cambridge, MA, USA. Scratch empowers users to create interactive stories, animations, games, and simulations [24]. Its drag-and-drop interface with coding blocks allows users to build and customize projects without prior programming experience. Scratch fosters creativity and logical thinking by enabling users to experiment with different programming concepts playfully and intuitively [25].

When integrated with Scratch (Figure 3), Makey Makey, opens up new possibilities for interactive projects. Scratch provides extensions explicitly designed for Makey Makey, enabling users to incorporate physical interactions into their digital creations. The combined use of Makey Makey and Scratch offers therapeutic benefits, such as fine motor skill refinement, creativity enhancement, and cognitive ability improvement.

Conductive materials like aluminum foil, conductive paints, and everyday objects such as fruits and play dough were employed for practical implementation. These materials were chosen for their ability to conduct electricity and their accessibility in educational settings. Connection cables with alligator clips were used to link these materials to the Makey Makey board, creating a flexible and adaptable input system. This setup supports the development of fine motor skills and provides a tangible and engaging way for children to interact with educational content.

Scratch, the visual programming language developed by MIT, located in Cambridge, MA, USA, was utilized to create the software component of the system. Scratch’s drag-and-drop interface allows for creating complex interactive applications without requiring advanced programming skills. This makes it an ideal tool for developing educational games and activities tailored to the needs of autistic children. The Scratch extensions for Makey Makey enable seamless integration of physical interactions into digital projects, enriching the user experience and enhancing the educational value of the activities [24,25]. The use of friendly colors and sensory elements ensures that the learning experience is both enjoyable and effective, catering to the unique needs of children with autism.

This integration fosters experiential learning, encourages exploration, and facilitates social interaction and collaboration. These tools offer a wide range of skill development opportunities, spanning task comprehension, active engagement, competitiveness, motivation, self-confidence, exposure to new stimuli (e.g., textures, sounds), and proficiency in various game categories [2].

## 4. Research Methodology: Theoretical and Practical Aspects

The implementation of play therapy offers a novel approach to supporting individuals diagnosed with autism spectrum disorders, aiming to complement conventional therapeutic methods. Embracing a holistic therapeutic approach, our team of software developers endeavor to utilize Makey Makey and Scratch to craft games explicitly tailored for support purposes. These games are intricately designed to encourage logical reasoning, facilitate the direct expression of emotions, augment emotional comprehension, promote empathy, and interactively refine memory and attention skills.

This initiative arises from recognizing that the challenges encountered by individuals with ASD can be broadly categorized in two realms: one centered on emotional and social development and the other focused on intellectual advancement. Extensive research underscores the hurdles faced by children with ASD in assimilating into society, ranging from establishing emotional connections to accomplishing tasks proficiently. By hamessing children’s enthusiasm for video games, this approach leverages enjoyment to stimulate motivation and active participation, ultimately fostering favorable behavioral changes [2].

### 4.1. Emotional Development

Research indicates that children with autism frequently encounter difficulties in emotional development that are often linked to anomalies in the limbic system, a critical component of the brain responsible for emotional behavior [4]. These anomalies can hinder their ability to form emotional connections, empathize with others, interpret facial expressions, and recognize their own emotions. Such challenges significantly affect their social interactions, leading to feelings of isolation, anxiety, and depression [5].

Interventions focusing on emotional and social skills are crucial for addressing these challenges. One innovative approach involves using games created with Makey Makey and Scratch, specifically designed to enhance emotional and social competencies. These games provide a structured and interactive platform for skill development, making the learning process engaging and effective.

Recent studies and practical applications of these tools have demonstrated positive outcomes. For example, the interactive nature of the games helps children with ASD practice and improve their ability to recognize and respond to various emotional cues in a controlled yet playful environment. By incorporating play therapy elements, these interventions make learning enjoyable and facilitate significant improvements in emotional comprehension and social interaction skills [2].

Results from these interventions have shown that children who engage with Makey Makey and Scratch games exhibit enhanced emotional recognition skills and empathy and an enhanced ability to navigate social situations. These improvements contribute to a higher quality of life by reducing feelings of isolation and anxiety and promoting better mental health and social integration.

#### 4.1.1. Emotion Recognition

The emotion recognition game helps autistic children identify and distinguish between four core emotions commonly experienced in daily life: anger, fear, happiness, and sadness. Before engaging in the game, children undergo a pre-test to evaluate their ability to identify emotions. This pre-test consists of a paper-based activity and a gamified self-assessment that associates expressions with events and emotions, incorporating tactile and software components utilizing Makey Makey and Scratch [2]. The pre-test aims to assess and reinforce the children’s existing knowledge, providing a comprehensive overview of the upcoming tasks (Figure 4).

The game setup involves placing the circuit in a colorful, child-friendly box with holes for all the involved wires (Figure 5). This setup is designed to integrate seamlessly into a floortime therapy session. The hardware component includes four templates representing the primary emotions, covered with aluminum foil on the back for conductivity and feature pockets for connecting cables. Each emotion template is linked to a specific keyboard key (up, down, left, and right arrows), and a ground connection is achieved by connecting a cable to the “earth” pin on the board and attaching the other end to a conductive accessory held by the child.

The game’s objective is to enhance children’s recognition of primary emotions. Children advance through eight scenarios, selecting the matching emotion template for each scenario. A cat character guides them through the game, presenting situations that depict primary emotions (Figure 6). Correct identification triggers a sound signal and a congratulatory message or applause, allowing the child to advance. Incorrect responses prompt encouragement for the child to try again.

Following the game’s completion, the children’s level of emotion recognition is re-assessed through a post-test (Figure 7). Unlike the pre-test, which uses drawings and emojis, the post-test presents expressions of real individuals, offering a more realistic measure of the children’s emotional identification abilities. This reassessment provides valuable insights into the children’s progress and highlights areas for further improvement.

This game aims to create a stress-free learning environment that fosters emotional awareness and understanding in autistic children. The game engages and educates children by utilizing interactive elements and real-life scenarios, ultimately empowering them to navigate social situations more effectively. This approach has shown promise in improving children’s ability to identify and understand emotions, thereby enhancing the overall quality of life for those with ASD.

#### 4.1.2. Social Story

The next game in the emotional development series focuses on fostering social interaction by enhancing understanding of social cues and expressions, while also developing observational skills, creativity, and empathy. Autism often presents significant communication and social engagement challenges, making it difficult for individuals to initiate and sustain connections, build friendships, and fully grasp the nuances of social norms and expectations [4]. Many individuals with autism struggle with interpreting non-verbal cues, participating in social activities, demonstrating empathy, and expressing themselves effectively. Some may even prefer solitude, engaging independently with objects or toys.

Set against the backdrop of a classic childhood outing—a school trip to a museum—this game unfolds in five captivating sequences. Each sequence invites players to decode images, unravel social cues, and piece together events in the correct order. As players progress, they place story cards onto the game board (Figure 8), marked with cheerful sun symbols and LEDs that light up upon correct placement.

Each space on the board features a unique design that is intricately linked to marks on the back of the story cards, completing an enchanting discovery circuit. To avoid confusion, all patterns are unique. With every step, LED lights illuminate in sync with specific keys on the Makey Makey board, and arrow keys and the SPACE key are used to navigate the journey (Figure 9).

As players delve deeper into the game, they are greeted with encouraging sounds and visuals, celebrating their successes and guiding them forward. The absence of repetition ensures clarity, allowing the narrative to shine seamlessly. Narrative consistency is maintained, and players can earn stars for each correct association (Figure 10).

The game aims to provide a structured yet engaging environment where children with autism can practice and enhance their social skills. The game simulates real-life social scenarios and helps children understand and interpret social cues, develop empathy, and improve their interaction abilities. The game’s interactive and rewarding nature also maintains the children’s interest and motivation, facilitating more effective learning and retention of social skills.

### 4.2. Attention and Memory Training

Children with autism spectrum disorder (ASD) frequently encounter challenges in maintaining attention due to sensory sensitivities, restricted interests, organizational difficulties, impulse control issues, social comprehension struggles, and transitional issues. These obstacles can significantly hinder their ability to focus on tasks or activities for extended periods. Adapting their environment and activities and providing personalized support can help address these attention-related difficulties specific to autistic children [6,26].

Various strategies and activities are employed to train attention and memory in children with autism, including attention games, concentration exercises, memory activities, interactive technology, and organizational strategies. These approaches aim to enhance attention, concentration, and memory skills through engaging and interactive methods, leveraging customized technology and structured environments to optimize learning experiences for children with autism [7,19].

Attention games and concentration exercises are designed to build the capacity for sustained focus incrementally. Memory activities often incorporate visual and auditory stimuli to reinforce learning and retention. Interactive technology, such as apps and software tailored to children with ASD, provides an engaging platform for these exercises. For example, Makey Makey and Scratch have shown promise in creating interactive environments where children can develop these skills in a fun and immersive way [2].

Organizational strategies, including visual schedules and structured routines, help children understand and predict activities, reducing anxiety and improving focus. By employing these multifaceted approaches, educators and therapists can create a supportive learning environment that addresses the unique needs of children with autism, enhancing their cognitive abilities and overall learning outcomes.

#### 4.2.1. Toy Hunt

Toy Hunt is an engaging game designed to enhance children’s attention and fine motor skills, particularly for those with autism spectrum disorder (ASD). The game challenges participants to extract four objects from a toy box using specially designed openings, encouraging precise hand and finger movements while avoiding excessive touches. With each successful extraction, participants are rewarded with sensory effects, including sound cues, visual prompts, and audio pronunciation of the identified object.

Although several games were considered, ‘Memory Maze’ was selected for the intervention because it is specifically designed to improve memory and concentration, two developmental areas that are particularly challenging for children with ASD. The other games were analyzed but were not included in the final intervention due to their focus on different cognitive and emotional areas.

The physical setup resembles a colorful toy box with four extraction zones designated for specific objects (Figure 11). Conductive materials, such as aluminum foil, cover the components, facilitating circuit completion when placed correctly on the board. The game incorporates gamification elements, with a buzzing sound accompanying any attempt to cross the forbidden boundary, aimed at regaining the child’s attention and motivating them to complete the task.

Key components of the setup include the use of keys on the Makey Makey board, with the SPACE key associated with touching the boundary, facilitating interaction and gameplay (Figure 12). Recognition relies on conductive materials, with aluminum foil covering the object parts for circuit completion. Connections between input pins, Makey Makey board keys, and objects are established, allowing interaction.

The game in Scratch includes random object extraction with no imposed order. Successfully placing an object triggers sensory effects, such as a sound, a green heart, and audio pronunciation. Touching the boundary prompts a more robust sound and the intermittent appearance of a red heart, designed to increase concentration (Figure 13). Errors are recorded for later observation of hand coordination improvement. The game continues until all objects are identified, with the final condition triggering a congratulatory window, showcasing variables and sensory effects. Through interactive gameplay and carefully crafted feedback mechanisms, Toy Hunt offers a supportive environment for children with ASD to practice attention and motor skills, fostering skill development and confidence.

#### 4.2.2. Memory Maze

The Memory Maze game is a valuable tool designed to enhance memory and concentration skills in children with autism spectrum disorder (ASD). Recent studies show that interventions aimed at improving memory and concentration can have a positive impact on cognitive development in children with ASD [14]. The initial description of the alternative games highlights the thorough analysis conducted before the final selection of the ‘Memory Maze’ game. Its clear and repetitive structure is particularly beneficial for individuals who thrive on routine, offering predictability and comfort while engaging them in cognitive tasks. Initially utilizing colored cards, the game now features colored clay buttons, providing a tactile sensation that stimulates interest and engagement.

Within the game, players are presented with sequences of colors and sounds, which gradually increase in complexity as they progress through levels. This progression challenges their memory skills and encourages sustained attention and focus. The physical setup initially included four colored cards with aluminum foil backs connected to corresponding arrow keys on the Makey Makey board. To enhance the tactile experience and make it more appealing to children, plasticine of the same color was used instead of cards (Figure 14).

The game begins with simple sequences that gradually increase in length and complexity. Players must remember and replicate these sequences using the colored clay buttons connected to the Makey Makey board. The conductive properties of the aluminum foil and clay ensure that each touch is registered accurately, facilitating smooth gameplay.

As the players advance, the sequences become more challenging, requiring greater concentration and memory retention. This incremental difficulty helps to build cognitive skills systematically. Sensory feedback, such as sound cues and visual prompts, reinforces learning and keeps the players engaged. The game’s repetitive nature also helps reinforce neural pathways, aiding in memory retention and recall.

While errors necessitate a restart, this mechanism encourages sustained attention and motivates players to strive for improvement, blending interactive gameplay with focused skill-building activities (Figure 15).

The Memory Maze game serves as an educational tool and provides therapeutic benefits by promoting mental stimulation and cognitive development. It is designed to create an enjoyable learning environment where children with ASD can improve their memory and concentration skills in a structured and supportive setting. Ultimately, the Memory Maze game provides a comprehensive and enjoyable avenue for enhancing these critical skills in autistic children.

## 5. Experimental Methodology, Results, and Discussion

### 5.1. Participants

An initial group of 60 children was selected for the study to evaluate the efficacy of the Memory Maze game in enhancing memory and concentration skills. The participants varied in age, gender, and cognitive abilities, providing a diverse sample for comprehensive analysis.

Participants in the study were recruited from local ASD therapy centers, consisting of 60 children aged 6–12 years who had been diagnosed according to DSM-5 criteria. The study was conducted in compliance with the Declaration of Helsinki and national legislation regarding research involving vulnerable populations. Parental consent was obtained for all participants via a written consent form which was thoroughly explained to parents or legal guardians. Children were informed that they could withdraw from the study at any point without any consequences, ensuring their autonomy and comfort. The assessments were conducted by licensed clinical psychologists, ensuring adherence to strict privacy and confidentiality protocols. Ethical approval was obtained from the relevant institutional review board, and all necessary measures were taken to safeguard the rights of the vulnerable participants.

The inclusion of ADHD assessments in this study was based on the high prevalence of comorbidity between ASD and ADHD, as documented in previous research. Given that children with ASD often exhibit symptoms of attention deficits and hyperactivity, it was essential to evaluate how these factors influenced the intervention’s effectiveness. Each participant’s ADHD diagnosis was confirmed using standardized clinical criteria from the DSM-5 (Diagnostic and Statistical Manual of Mental Disorders, Fifth Edition) and administered by licensed psychologists. The sample included children with the following ADHD subtypes: Inattentive, Hyperactive–Impulsive, and Combined. ADHD scores were collected and analyzed to assess their impact on the intervention, with results indicating significant variations in response based on ADHD subtype.

The participants in this study were randomly selected from local ASD therapy centers. The sample consisted of 60 children, aged between 6 and 12 years (with a mean age of 9 years), representing a demographic group from Southeastern Europe. Of the participants, 55% were boys and 45% were girls. Before the intervention, each child underwent standardized cognitive tasks that are commonly used for evaluating memory and concentration in children with ASD. These tasks were adapted from established instruments, including the Wechsler Intelligence Scale for Children (WISC) and the Conners’ Continuous Performance Test (CPT). The WISC was used to assess general cognitive functioning, while the CPT measured attention and impulse control. All assessments were conducted by licensed clinical psychologists specializing in developmental disorders, ensuring that the tests were administered in a consistent and reliable manner.

The experimental procedure involved several key phases:Pre-Test Assessment: Each child underwent a pre-test assessment to establish a baseline for their memory and concentration abilities. The pre-test consisted of cognitive tasks designed to measure initial performance, with possible scores ranging from 0 to 100 points. The group of children achieved scores ranging from 40 to 69 points. Factors such as age, gender, and initial cognitive scores were recorded.Intervention with Memory Maze: The children were introduced to the game. This interactive tool was employed over a specified period, during which children engaged with the game in a structured environment. The game presented sequences of colors and sounds, increasing in complexity to challenge the participants’ memory and attention skills progressively.Game Setup: The physical setup of the game included colored clay buttons connected to a Makey Makey board. Each button represented a color and was covered with conductive materials to facilitate accurate interaction. The game’s interface, developed in Scratch, provided immediate sensory feedback (visual and auditory) to enhance engagement and reinforce learning. The pre-test scores ranged from 40 to 69 points (with a mean score of 54.5 points), reflecting a balanced distribution of initial cognitive abilities among the participants. These data provide a solid basis for evaluating the intervention’s effectiveness by measuring improvements in the children’s memory and concentration following the use of the ‘Memory Maze’ game.Post-Test Assessment: Following the intervention period, the children were re-assessed using a post-test identical in structure to the pre-test. This assessment measured any improvements in memory and concentration, with final scores recorded for each participant.

### 5.2. Data Collection

The primary collected data included age: the participants ranged from 6 to 12 years. The group included both male and female participants. Baseline cognitive performance scores were obtained from a pre-test. Cognitive performance scores were obtained from the post-test after the Memory Maze game testing phase. All procedures performed in this study involving human participants were in accordance with the ethical standards of the institutional and national research committee and the 1964 Helsinki Declaration and its later amendments, or comparable ethical standards.

### 5.3. ADHD Diagnosis

The study participants were diagnosed with ADHD through a comprehensive evaluation process conducted by licensed mental health professionals, typically psychologists, psychiatrists, or pediatricians. The diagnosis of ADHD is based on standardized criteria outlined in the Diagnostic and Statistical Manual of Mental Disorders, Fifth Edition (DSM-5), published by the American Psychiatric Association [1]. The diagnostic process includes clinical interviews with parents, teachers, and the child, depending on their age and communication abilities. Information about the child’s behavior at home and in school is gathered through these structured interviews [26]. Behavior rating scales and checklists, such as the Conners’ Rating Scales, Vanderbilt ADHD Diagnostic Rating Scales, and the ADHD Rating Scale-IV, are commonly used to assess the frequency and severity of ADHD symptoms [27,28]. Clinicians may also observe the child in different settings, such as at home or in school, to see how they behave in various environments [29]. Additionally, cognitive and neuropsychological tests, such as the Continuous Performance Test (CPT) and the Wechsler Intelligence Scale for Children (WISC), help identify attention and executive function deficits [30].

There are three main types of ADHD, classified based on their predominant symptoms: ADHD, predominantly Inattentive type; ADHD, predominantly Hyperactive–Impulsive type; and ADHD, Combined type [1]. The type of ADHD can influence the approach to intervention and the specific areas of improvement targeted by the Memory Maze game. Children with predominantly inattentive symptoms may benefit more from tasks designed to enhance sustained attention. In contrast, those with hyperactive–impulsive symptoms might require activities that promote impulse control and executive functioning. Understanding the type of ADHD helps in tailoring interventions to meet each child’s specific needs, thereby optimizing therapeutic outcomes [1].

### 5.4. Methods Ensuring Accurate Test Results

Several methodological approaches were employed to ensure the accuracy and reliability of the test results. A pre-test and post-test design allowed for a clear comparison of cognitive abilities before and after the intervention, helping to measure the direct impact of the Memory Maze game on the participants’ memory and attention skills. The mental tasks used in the pre-test and post-test were standardized, ensuring consistency in what was being measured, with scores ranging from 0 to 100 points. All testing was conducted in a controlled environment to reduce the influence of external factors on the participants’ performance, including by maintaining consistent lighting, noise levels, and testing materials.

The sequences of colors and sounds in the Memory Maze game were randomized to prevent learning effects from previous exposures, ensuring scores were improved due to enhanced cognitive abilities rather than the memorization of specific sequences. Participants received exact instructions and guidance throughout the testing process, helping to eliminate variability in how the children were presented with and understood the tasks. Observers and facilitators involved in the testing were thoroughly trained to administer the tests and interact with participants consistently, including handling the Makey Makey setup and providing instructions without influencing the children’s responses.

Data were meticulously recorded and analyzed to identify patterns and ensure accuracy. Statistical analysis tools were used to assess the significance of improvements and control for any potential biases in the data. Errors made during the tasks were carefully tracked and analyzed, providing insight into areas where participants struggled and helping to refine the tasks and instructions for clarity and effectiveness. By implementing these rigorous methods, the study aimed to ensure that the test results accurately reflected the participants’ improvements in memory and concentration due to the Memory Maze game.

### 5.5. Statistics

A paired *t*-test was conducted to evaluate the significance of the improvements in scores following the intervention with the Memory Maze game. This test is appropriate for comparing the means of two related groups: the initial and final scores of the same participants. The paired *t*-test assesses whether the mean difference between these two scores is statistically significant.

The paired *t*-test was chosen because it considers that the same children were tested before and after the intervention, thus controlling for individual differences in performance. By comparing the initial and final scores, the test provides a robust measure of the effectiveness of the Memory Maze game in improving memory and concentration skills.

In addition to the paired *t*-test, descriptive statistics such as the mean, standard deviation, minimum, maximum, and percentiles were calculated to provide a comprehensive overview of the data. These statistical measures helped to quantify overall improvement and identify trends across different age groups and genders.

### 5.6. Results

Below is a summary table, Table 1, of the data grouped by age, showing the mean initial scores, final scores, and improvements for both females and males:

This table provides a detailed overview of the performance and improvements of boys and girls across different age groups after using the Memory Maze game.

The differences between the initial and final scores were analyzed to determine the effectiveness of the Memory Maze game. Improvements were calculated and categorized by age and gender to identify trends or patterns. Statistical analysis provided insights into the average improvement across different demographics, highlighting the game’s impact on enhancing memory and concentration skills in autistic children.

The table below, Table 2, summarizes the central results by age, including the mean initial score, final score, and improvement:

The scores presented in this table refer to children’s performance on a specific set of cognitive tasks designed to assess their memory and concentration skills. The initial score represents the children’s performance on these tasks before using the Memory Maze game. The tasks involve sequences of colors and sounds that users must remember and replicate. The final score represents the children’s performance on tasks that are either the same or very similar to those used in the pre-test after using the Memory Maze game. It measures the improvements in their memory and concentration skills. The difference between the initial and final scores indicates an improvement in children’s cognitive skills after engaging with the Memory Maze game.

The scoring scale for the Memory Maze game ranges from 0 to 100 points, with higher scores indicating better performance on the cognitive tasks. The scale is categorized into four performance levels. Scores between 0 and 30 points indicate poor performance, where users struggle to maintain attention and memory. Scores between 31 and 60 points represent average performance, indicating that users demonstrate moderate abilities in attention and memory but still have room for improvement. Scores between 61 and 90 points signify good performance, indicating that users have strong attention and memory skills. Scores between 91 and 100 points reflect excellent performance, demonstrating that users have high attention and memory skills.

In addition to the *t*-test, an Analysis of Variance (ANOVA) was conducted to evaluate the influence of ADHD subtype (Inattentive, Hyperactive–Impulsive, Combined) and gender (male, female) on the improvement in memory and concentration scores. The dependent variable was the improvement in scores, defined as the difference between the pre-test and post-test scores.

The results of the ANOVA indicated that ADHD subtype had a significant effect on the improvement in memory and concentration scores (F(2, 57) = 5.34, *p* < 0.01). Specifically, children with the Hyperactive–Impulsive subtype showed the highest average improvement, with a mean increase of 15 points in memory and concentration scores, compared to 10 points for the Inattentive subtype and 8 points for the Combined subtype.

In contrast, gender did not have a significant effect on the improvement in scores (F(1, 58) = 1.23, *p* > 0.05), indicating that both male and female participants exhibited similar responses to the intervention. These results suggest that ADHD subtype plays a crucial role in determining the effectiveness of the intervention, while gender does not appear to be a significant factor.

To ensure the statistical validity of the results, a power analysis was conducted to determine whether the sample size of 60 participants was sufficient. The analysis indicated that, with an effect size of 0.5 and a significance level of *p* < 0.05, the sample size provided a statistical power of 0.80. This level of power is generally considered sufficient to detect medium-sized effects in intervention studies. As such, the sample size was deemed appropriate for the purposes of this study and adequate for ensuring the validity of the results.

### 5.7. Statistical Analysis and Results Based on ADHD Type

Furthermore, to analyze the effectiveness of the Memory Maze game, participants were categorized based on their type of ADHD: Inattentive, Hyperactive–Impulsive, and Combined.

### 5.8. Discussion

The analysis of improvements in scores following intervention with the Memory Maze game reveals significant gains in memory and concentration skills among the participants. Overall, the mean improvement across all children was 23.38 points, with a standard deviation of 4.11 points, indicating a consistent enhancement in performance. The minimum observed improvement was 15 points, while the maximum was 29 points. The 25th percentile of improvement was 20.75 points, the median (50th percentile) was 24 points, and the 75th percentile was 27 points.

When analyzing improvements by gender, female participants showed an average improvement of 24.16 points, while male participants exhibited a slightly lower average improvement of 23.02 points.

Improvements were also analyzed by age group, revealing the following average improvements: children aged 6 demonstrated an average improvement of 24.50 points; children aged 7 showed an average improvement of 20.83 points; children aged 8 had an average improvement of 22.60 points; children aged 9 exhibited an average improvement of 22.50 points; children aged 10 showed the highest average improvement of 25.67 points; children aged 11 demonstrated an average improvement of 23.57 points; and children aged 12 exhibited an average improvement of 23.90 points.

These results indicate that the Memory Maze game effectively enhances cognitive skills across different ages and genders, with particularly notable improvements observed in younger children and those aged 10. The game’s structured, interactive nature and its repetitive and engaging tasks contribute to these significant cognitive gains.

After analyzing the statistical analysis and the results based on ADHD type in Table 3, we observed that all three groups significantly improved their scores after using the Memory Maze game. The mean improvement for children with the Inattentive type of ADHD was 23.53 points; for those with the Hyperactive–Impulsive type, it was 25.00 points; and for those with the Combined type, it was 24.15 points. These improvements highlight the effectiveness of the Memory Maze game across different types of ADHD, with the Hyperactive–Impulsive group showing the highest average improvement.

When conducting an ANOVA to assess the influence of gender and age on the intervention outcomes, we found the following results.

Gender did not have a statistically significant impact on the results (F(1, 58) = 1.23, *p* > 0.05), indicating that both boys and girls benefited similarly from the intervention. Slight differences in improvement were observed, with boys showing marginally higher improvements than girls, but these differences were not statistically significant. These findings suggest that, while minor variations may exist due to factors like engagement or learning style, the intervention was equally effective across genders.

Age had a more pronounced effect on the outcomes (F(2, 57) = 4.15, *p* < 0.05). Children aged between 8 and 10 demonstrated the greatest improvements in their scores on memory tasks compared to younger and older children. This likely reflects the cognitive development stage of children in this age group, who may be more receptive to memory and concentration-focused interventions. These results suggest that interventions may need to be tailored according to developmental stages to maximize their effectiveness.

The results demonstrated significant improvements in both memory and concentration following the use of the Memory Maze game, confirming the effectiveness of the intervention.

A Technology Acceptance Model (TAM) was used to evaluate the acceptance of the Memory Maze game by children with ASD and their caregivers. The TAM measures two key factors that influence user acceptance: Perceived Usefulness (PU) and Perceived Ease of Use (PEOU). PU refers to the degree to which the game is seen as helpful in improving memory and concentration, while PEOU measures how simple and intuitive the game is to use. In our evaluation, both children and caregivers rated the game highly on both PU and PEOU, indicating that the game was perceived as useful and easy to use. These findings suggest that the Memory Maze game has the potential to be widely accepted as a tool for cognitive improvement in children with ASD. Additionally, we collected feedback on the Attitude Toward Using and Behavioral Intention to Use, which showed positive responses, confirming the potential for the regular use of the game.

Our findings are consistent with those of Zeidan et al. [2], who reported global prevalence rates of autism and emphasized the importance of early intervention strategies, such as those applied in our study. Additionally, Budisteanu et al. [3] conducted a prevalence study in Romania, providing context for our sample and further underlining the necessity of localized interventions targeting cognitive skills in children with ASD.

Our results also align with Levy et al. [4], who highlighted the critical role of cognitive interventions in improving outcomes for children with ASD. Similar to our work, their study found that structured, game-based approaches significantly improved memory and concentration. Moreover, Dechsling et al. [19] and Ip et al. [20] explored the application of virtual reality in ASD interventions, focusing on social and emotional skills. While our study primarily addressed cognitive functions, the success of these virtual reality-based interventions supports the effectiveness of interactive digital tools in promoting broader developmental outcomes.

Overall, the Memory Maze game effectively enhanced memory and concentration skills in children with different types of ADHD. The significant improvements observed across all groups suggest that this intervention can be broadly applied to support children with varying ADHD presentations, contributing positively to their cognitive development and therapeutic outcomes [1,26,27,28,29,30].

The Memory Maze game’s structured and supportive setting, combined with its interactive and repetitive nature, proved it to be an effective method for cognitive enhancement. The findings demonstrated significant improvements in the children’s ability to maintain attention and enhancements in their memory, underscoring the potential of such gamified interventions in educational and therapeutic contexts.

Considering several demographic factors, our comprehensive approach ensured robust and reliable results, paving the way for future research and development in similar interventions. This study highlights the importance of using tailored, interactive tools like the Memory Maze game to support cognitive development in children with ASD, ultimately contributing to their overall well-being and quality of life.

## 6. Limitations

Although the sample size of 60 participants was adequate for the statistical analysis performed in this study, it may limit the broader generalizability of the findings. A larger sample size in future research would provide more robust data to confirm these results and extend their applicability to a wider population. Additionally, the sample was limited to children from a specific geographic region, potentially restricting the applicability of the findings to other populations. Furthermore, the short duration of the intervention may affect the long-term impact of the game on memory and concentration. Future research should address these limitations by expanding the sample size and geographic scope, and by evaluating the long-term effects of the intervention.

## 7. Conclusions

Developing and implementing therapeutic games for children with autism utilizing Makey Makey and Scratch have revealed several significant findings. These games represent a comprehensive approach to therapy, addressing key areas such as concentration, motivation, social interaction, experiential learning, and academic resources. By integrating physical components, these games offer a unique advantage over online-only applications, establishing tangible connections to personal objects and fostering real-life connections within therapy sessions.

The data indicate that, on average, children showed significant improvement in their memory and concentration skills after using the Memory Maze game. The mean improvement across all age groups was approximately 23.38 points, with the most notable enhancements observed in children aged 10, who showed an average improvement of 25.67 points. Females exhibited a slightly higher average improvement than males, with 24.16 points and 23.02 points, respectively.

The differences between the initial and final scores were analyzed to determine the effectiveness of the Memory Maze game. Improvements were calculated and categorized by age and gender to identify trends or patterns. Statistical analysis provided insights into the average improvements across different demographics, highlighting the game’s impact on enhancing memory and concentration skills in autistic children.

The Memory Maze game demonstrates significant potential to enhance memory and concentration skills in children with ADHD. Across all three types of ADHD—Inattentive, Hyperactive–Impulsive, and Combined—participants showed substantial improvements in their cognitive abilities. The game’s tailored approach allows it to address the specific needs of each ADHD type, promoting sustained attention, impulse control, and overall executive functioning.

The data indicate that the Memory Maze game facilitates cognitive skill development and engages children interactively and enjoyably. The game supports a wide range of therapeutic outcomes by providing a structured yet flexible learning environment, making it a valuable addition to existing ADHD interventions.

Also, the results of the ANOVA analysis demonstrated that ADHD subtype had a significant effect on the improvement in memory and concentration scores, with children diagnosed with the Hyperactive–Impulsive subtype showing the greatest improvements. Gender, on the other hand, did not significantly influence the intervention outcomes. These findings highlight the importance of considering individual characteristics, such as ADHD subtype, when designing interventions aimed at improving cognitive functions in children with ASD.

A pivotal aspect of these games is their thoughtful design, which incorporates friendly colors, sensory comfort, and the absence of limiting factors such as limited lives. These design considerations contribute to a positive and engaging experience for children with autism, enhancing their overall participation and enjoyment. Furthermore, including affective reinforcement mechanisms, such as sound cues and encouraging messages, plays a vital role in sustaining motivation throughout gameplay.

The repeatable nature of these games allows for revisits without additional effort, facilitating enhanced learning and reducing the need for extrinsic rewards. Additionally, incorporating quantification and testing features enables a detailed assessment of progress, facilitating the customization of interventions to meet individual needs and optimizing therapy outcomes.

The Memory Maze game was well received by both children with ASD and their caregivers, as shown by the results of the Technology Acceptance Model (TAM) analysis. The game was rated highly for Perceived Usefulness (PU) and Perceived Ease of Use (PEOU), suggesting strong potential for its adoption as a cognitive improvement tool. Additionally, positive feedback on Attitude Toward Using and Behavioral Intention to Use further supports the game’s potential for regular use.

We observed an immense potential for further improvements in therapeutic gaming for children with autism. Areas for future development include direct training approaches, enhanced monitoring capabilities, the incorporation of progressively challenging levels, improved physical components, and the creation of new applications tailored to address diverse challenges faced by children with autism and other disabilities. By continually refining and expanding these therapeutic tools, we can better support the unique needs and abilities of autistic children, ultimately improving their quality of life and fostering their overall development. In the future, we aim to test the Memory Maze game across various demographics to understand its effectiveness better. Additionally, we plan to broaden its use to cater to other conditions such as ADHD, learning disabilities, and anxiety disorders.

Future research will focus on several key areas: conducting longitudinal studies to assess the long-term impact of the Memory Maze game on cognitive and social development in children with autism and other conditions; incorporating customization features that allow the game to be tailored to the specific needs and abilities of each child; engaging parents and teachers in the therapeutic process by providing them with tools and resources to support the child’s development at home and in school; exploring the integration of the Memory Maze game with other therapeutic approaches, such as speech therapy, occupational therapy, and behavioral interventions; utilizing advanced data analytics to gain deeper insights into the effectiveness of the game; developing mobile and online versions of the Memory Maze game to increase accessibility and allow children to engage with the game outside of traditional therapy sessions; adding multilingual support to the game to cater to children from diverse linguistic backgrounds; and introducing collaborative gameplay features that allow children to play and interact with their peers.

Integrating advanced data analytics, customization features, and parent–teacher engagement strategies will also enhance the therapeutic potential of this innovative tool. Overall, the Memory Maze game offers a promising avenue for improving quality of life and developmental outcomes for children with ADHD, supporting their cognitive growth and academic success.

## Figures and Tables

**Figure 1 sensors-24-06720-f001:**
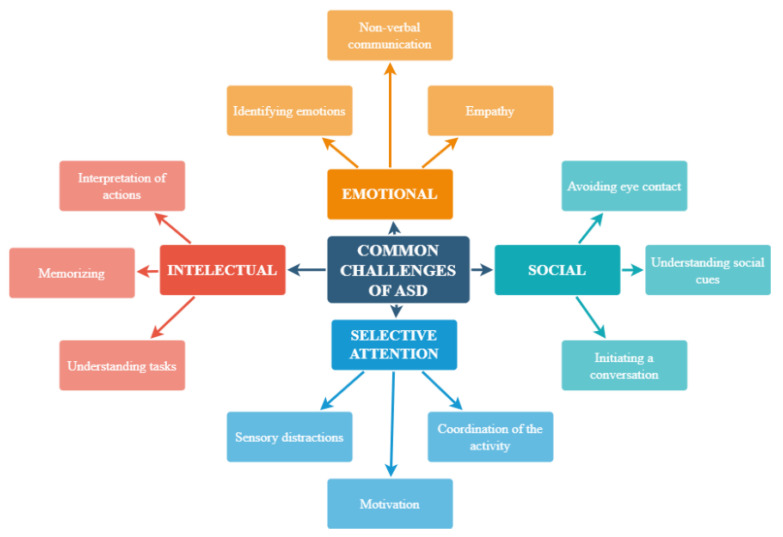
Common challenges in ASD.

**Figure 2 sensors-24-06720-f002:**
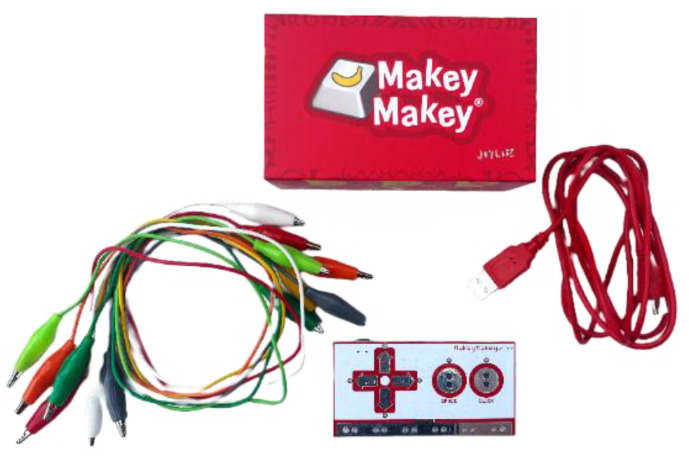
Makey Makey kit components.

**Figure 3 sensors-24-06720-f003:**
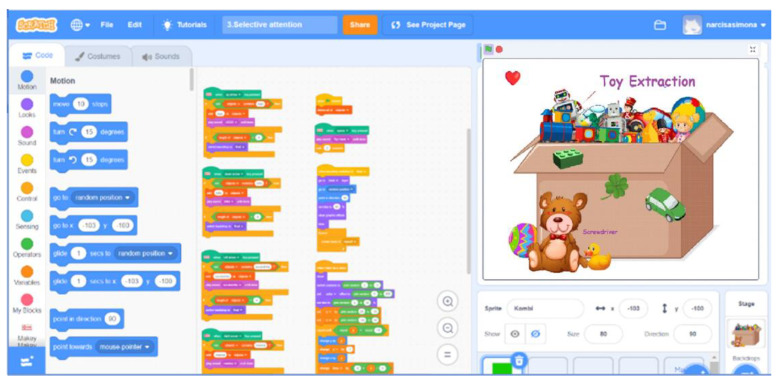
Scratch—coding environment.

**Figure 4 sensors-24-06720-f004:**
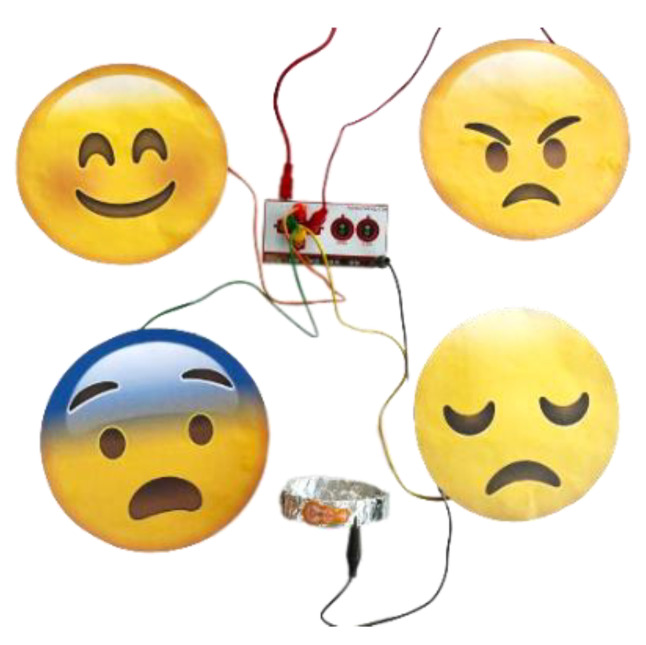
The physical setup corresponding to the pre-test.

**Figure 5 sensors-24-06720-f005:**
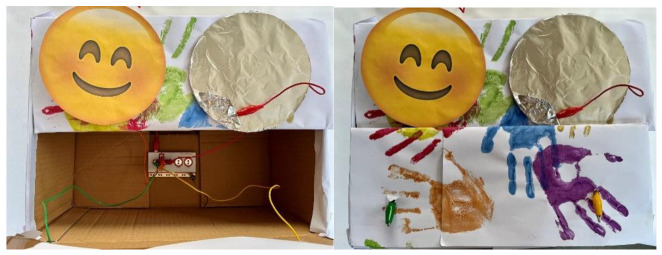
Game interface environment.

**Figure 6 sensors-24-06720-f006:**
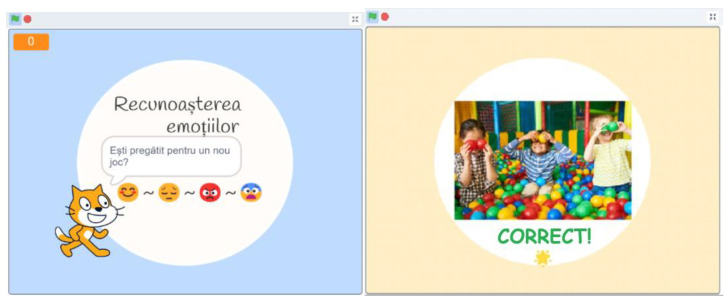
Game interfaces.

**Figure 7 sensors-24-06720-f007:**
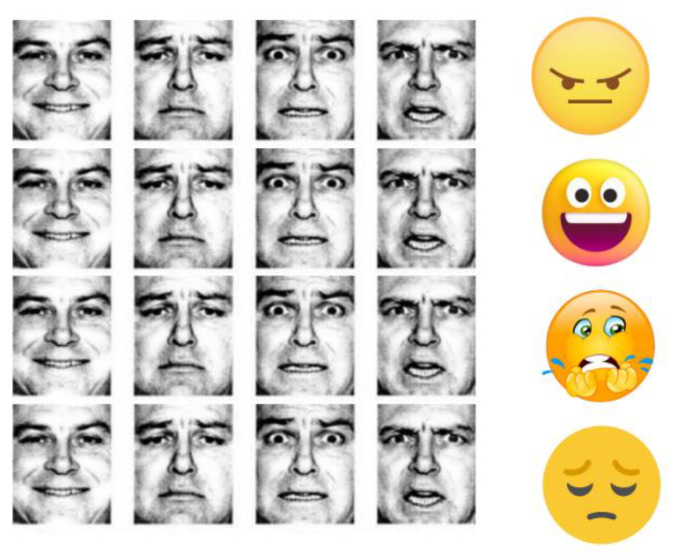
Post-test experiment.

**Figure 8 sensors-24-06720-f008:**
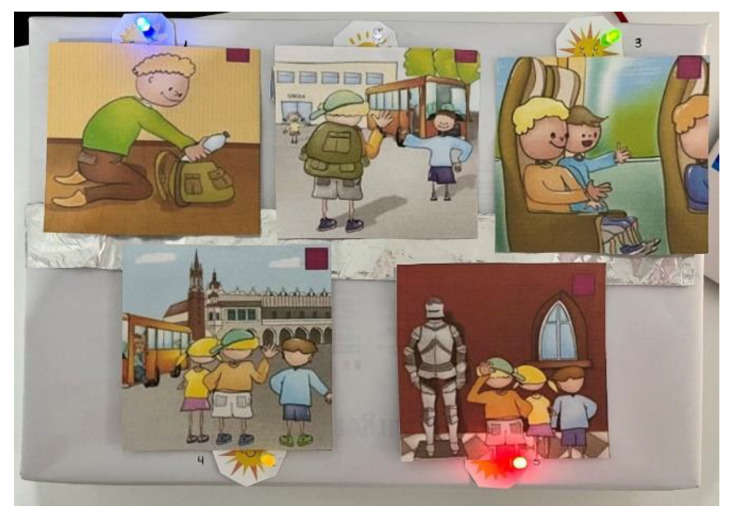
Social storyboard.

**Figure 9 sensors-24-06720-f009:**
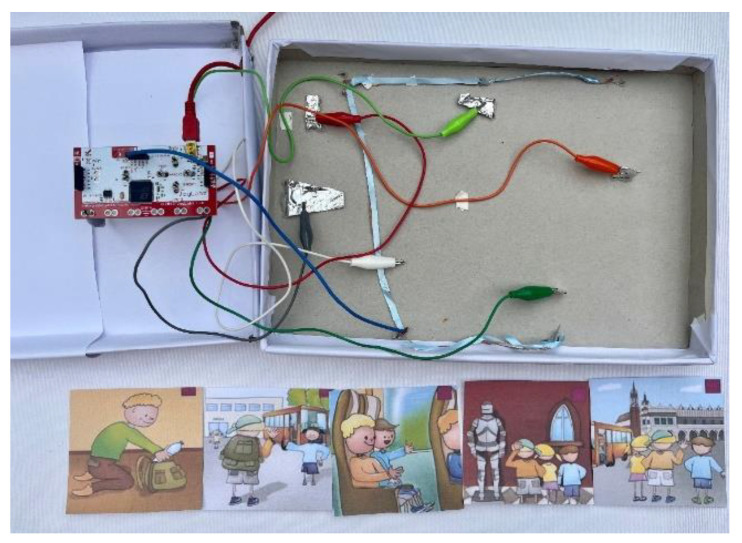
Initial setup with Makey Makey.

**Figure 10 sensors-24-06720-f010:**
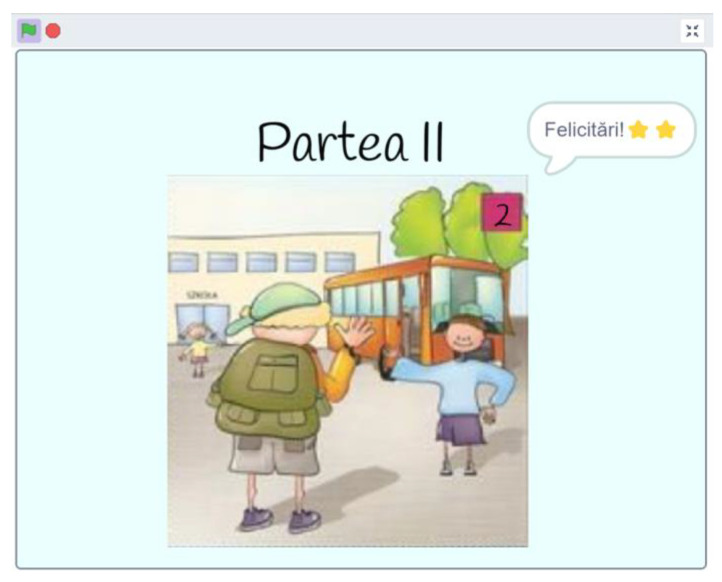
Scratch interface corresponding to winning two stars.

**Figure 11 sensors-24-06720-f011:**
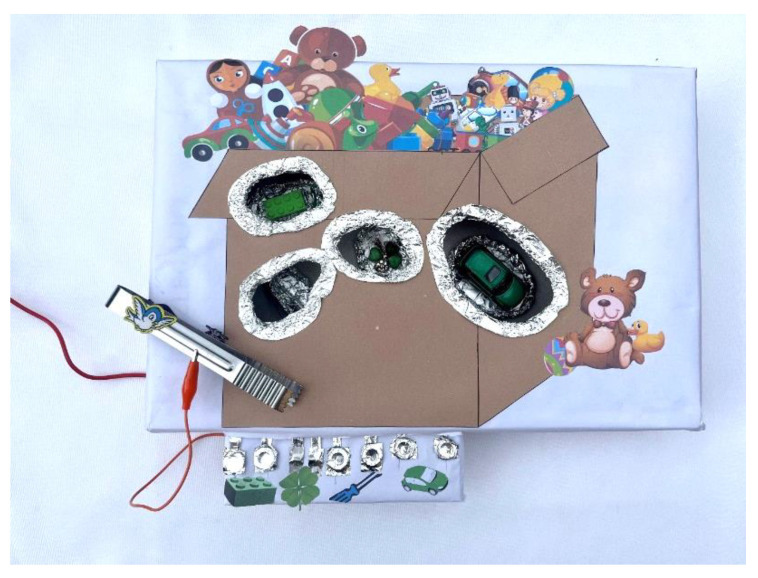
Physical setup of the extraction game.

**Figure 12 sensors-24-06720-f012:**
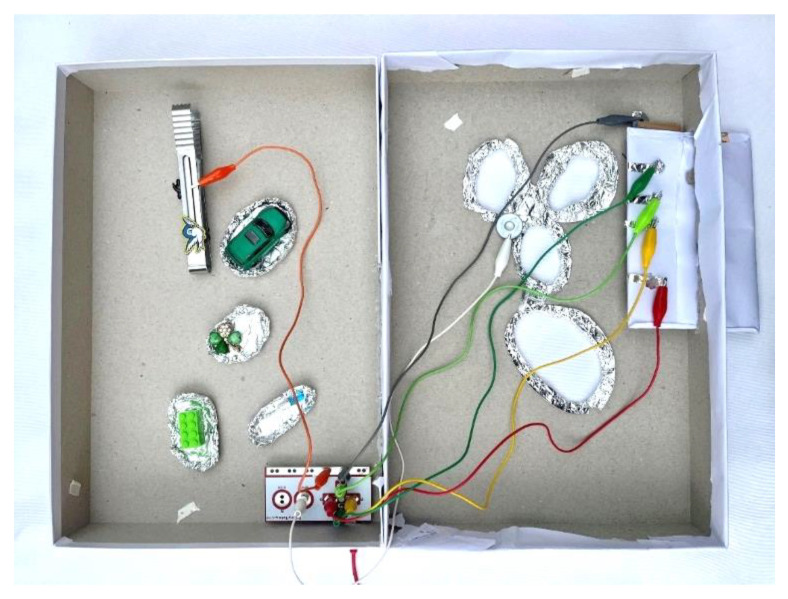
Makey Makey board connections.

**Figure 13 sensors-24-06720-f013:**
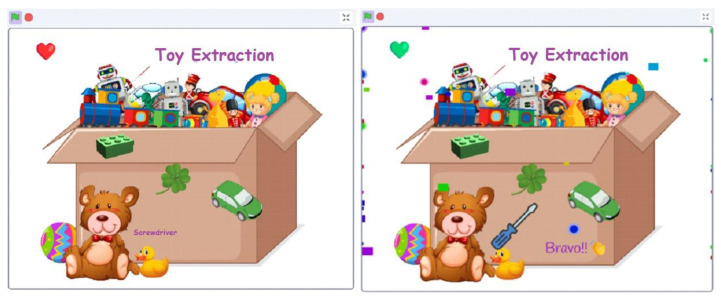
Interface in Scratch: correct positioning of all objects (**right**), wrong positioning (**left**).

**Figure 14 sensors-24-06720-f014:**
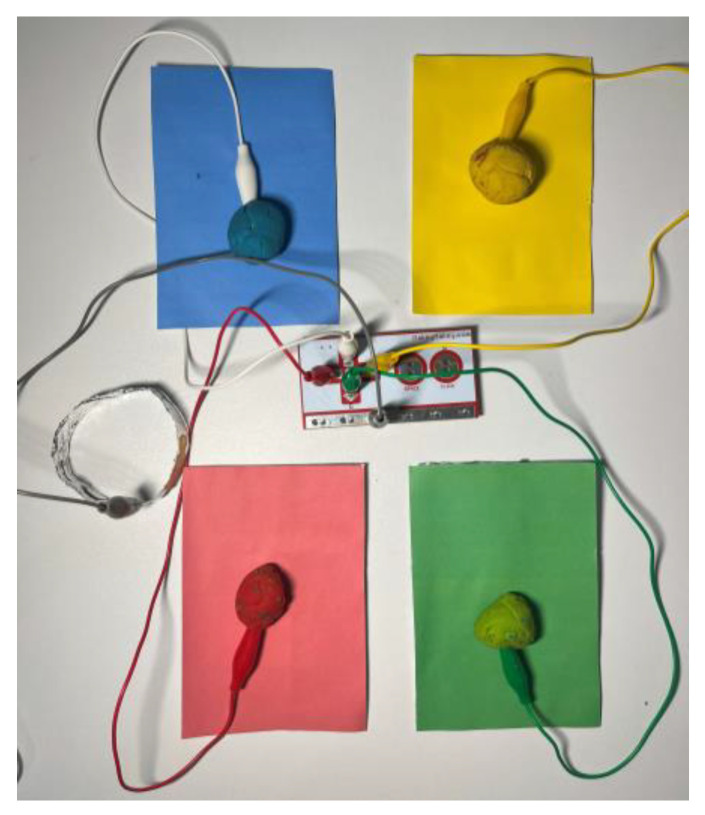
Physical setup of the game.

**Figure 15 sensors-24-06720-f015:**
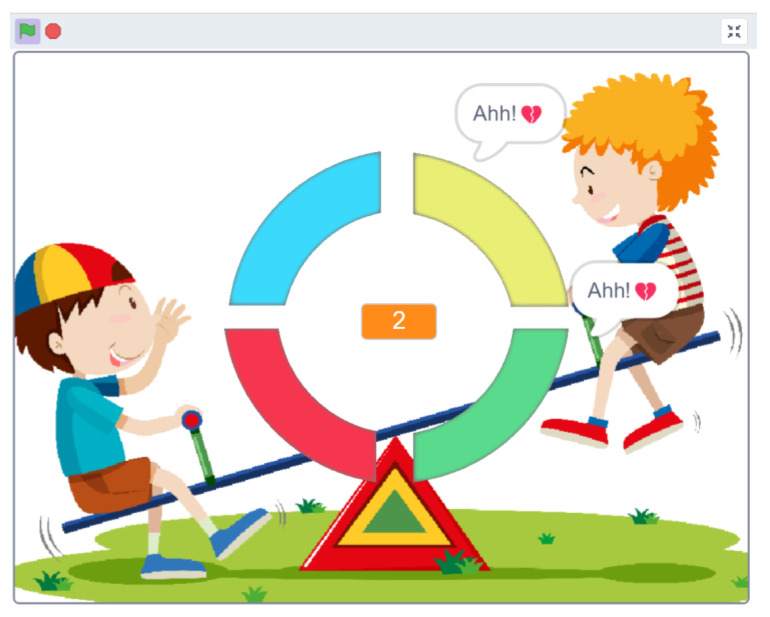
Interface corresponding to the moment of loss.

**Table 1 sensors-24-06720-t001:** Mean Initial and Final Scores, and Improvements for Females and Males by Age Group.

Age	Female Initial Score (Mean)	Female Final Score (Mean)	Female Improvement (Mean)	Male Initial Score (Mean)	Male Final Score (Mean)	Male Improvement (Mean)
6	43.50	68.75	25.25	48.00	71.00	23.00
7	58.00	79.20	21.20	57.20	77.80	20.60
8	52.67	75.00	22.33	50.33	73.33	23.00
9	55.33	77.67	22.33	53.33	76.33	23.00
10	52.25	78.75	26.50	50.67	75.67	25.00
11	47.00	72.00	25.00	49.67	72.67	23.00
12	53.00	77.00	24.00	50.67	74.33	23.67

**Table 2 sensors-24-06720-t002:** Central results of children by age, including the mean initial score, final score, and improvement.

Age	Gender	Initial Score (Mean)	Final Score (Mean)	Improvement (Mean)
6	Mixed	46.17	70.67	24.50
7	Mixed	57.50	78.33	20.83
8	Mixed	51.50	74.10	22.60
9	Mixed	54.50	77.00	22.50
10	Mixed	51.56	77.22	25.67
11	Mixed	48.57	72.14	23.57
12	Mixed	51.70	75.60	23.90

**Table 3 sensors-24-06720-t003:** The mean initial scores, final scores, and improvements for each ADHD type.

Adhd Type	Initial Score (Mean)	Final Score (Mean)	Improvement (Mean)	Initial Score (Std)	Final Score (Std)	Improvement (Std)
Inattentive	51.68	75.21	23.53	5.83	6.17	4.39
Hyperactive–Impulsive	50.67	75.67	25.00	5.49	5.83	4.00
Combined	51.11	75.26	24.15	5.82	6.15	4.12

## Data Availability

The original contributions presented in the study are included in the article. Further inquiries can be directed to the corresponding authors.

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
