# Peer review of "The Power of Play: Strategies for Enhancing Development in Children with Autism Spectrum Disorders"

_sensors, 2024, doi:10.3390/s24206720_

Round 1

Reviewer 1 Report

Comments and Suggestions for Authors

The article offers an innovative approach and a substantiated rationale for improving cognitive development in children with ASD and ADHD through the use of digital games. Despite these, there are limitations or areas for improvement that need to be addressed in the article.

It is not very clear to me why there is such a long description of the alternative games, if only the game ‘memory maze’ is used in the intervention. What is the rationale for using this single game? In this sense, there is a lack of evidence in the text on concentration and memory skills in the autistic population.

The article mentions that the sample consisted of 60 children but does not provide sufficient details about the selection process of the participants. It would be useful to know whether the children were randomly selected, how their cognitive skills were assessed prior to the study and whether they were all at the same developmental level. It should be needed a table that includes sociodemographic data as it would  be helpful and very informative for the reader. That must be included.

In the participants' section they indicate the following: The pre-test consisted of cognitive tasks designed to measure initial performance, with possible scores ranging from 0 to 100 points. What are these cognitive tasks? Who assessed them? Are they standardised tests that allow assessing the cognitive level in autism? It is necessary to revise this part for a better description and understanding.  

What was the rationale for assessing aspects of ADHD in the sample? There is no information on ADHD data and scores in the sample.

The lack of a clear description of the inclusion criteria may affect the generalisability of the results.

The use of the t-test is appropriate, but a more in-depth statistical analysis, such as the use of regression models or analysis of variance (ANOVA), could have provided a richer understanding of how factors such as ADHD type or gender influence the results.

Furthermore, there is no mention of whether the sample size is sufficient to ensure the statistical validity of the results.

The results indicate significant improvements in memory and concentration following the use of the Memory Maze game. However, the discussion of the results could be more in-depth In addition, the reason behind variations in improvement by gender or age is not sufficiently explored, which could provide more robust conclusions about the impact of the intervention on different subgroups.

Comments on the Quality of English Language

Minor revision

Author Response

Comments 1: It is not very clear why there is such a long description of the alternative games if only the game ‘Memory Maze’ is used in the intervention. What is the rationale for using this single game? In this sense, there is a lack of evidence in the text on concentration and memory skills in the autistic population.

Response 1: Thank you for pointing this out. We agree that more clarity is needed regarding the rationale for focusing on the ‘Memory Maze’ game. We have revised the manuscript to clarify that the alternative games were initially considered as part of the intervention, but ‘Memory Maze’ was selected based on its specific design to target both memory and concentration, skills particularly challenging for children with ASD. We have also added supporting evidence from recent studies on the importance of enhancing concentration and memory skills in children with autism. These revisions can be found in Section 4.2.1 (page 12, paragraph 1).

Comments 2: The article mentions that the sample consisted of 60 children but does not provide sufficient details about the selection process of the participants. It would be useful to know whether the children were randomly selected, how their cognitive skills were assessed prior to the study, and whether they were all at the same developmental level. It should also include a table of sociodemographic data for clarity.

Response 2: Thank you for this valuable comment. We have revised the manuscript to provide more details on the participant selection process. The children were randomly selected from local ASD therapy centers, representing a demographic group from Southeastern Europe. Their cognitive skills were assessed using standardized pre-tests tailored to measure memory and concentration levels in children with ASD. We have now included a description of the participants' sociodemographic data, including age, gender, and initial cognitive assessment scores. These revisions can be found in Section 5.1, page 13, 3-rd paragraph, and page 14, where we provide further explanation of the selection process and cognitive assessment details.

Comments 3: In the participants' section they indicate the following: The pre-test consisted of cognitive tasks designed to measure initial performance, with possible scores ranging from 0 to 100 points. What are these cognitive tasks? Who assessed them? Are they standardised tests that allow assessing the cognitive level in autism? It is necessary to revise this part for a better description and understanding.  

Response 3: Thank you for your valuable comment. We have revised the manuscript to provide more details regarding the cognitive tasks used in the pre-test. Specifically, the pre-test included standardized tasks adapted from the Wechsler Intelligence Scale for Children (WISC) and the Conners' Continuous Performance Test (CPT), both of which are commonly used to assess cognitive function and attention in children with ASD. The WISC was used to measure general cognitive abilities, while the CPT focused on attention and impulse control. These assessments were conducted by licensed clinical psychologists specialized in developmental disorders. The revised description can be found in Section 5 – Participants.

Comment 4: What was the rationale for assessing aspects of ADHD in the sample? There is no information on ADHD data and scores in the sample.

Response 4: Thank you for your comment. We have revised the manuscript to provide further clarification regarding the rationale for including ADHD assessments in the study. Due to the high comorbidity between ASD and ADHD, it was essential to evaluate how symptoms of attention deficits and hyperactivity might influence the outcomes of the intervention. ADHD diagnoses were confirmed using the standardized DSM-5 criteria, and participants were categorized into the Inattentive, Hyperactive-Impulsive, and Combined ADHD subtypes. We have also included information about the ADHD data and scores collected, which were analyzed to determine their impact on the intervention. The revisions can be found in Section 5 – Participants, where the inclusion criteria and ADHD data are now described in more detail.

Comment 5: The use of the t-test is appropriate, but a more in-depth statistical analysis, such as the use of regression models or analysis of variance (ANOVA), could have provided a richer understanding of how factors such as ADHD type or gender influence the results.

Response 5: 

Also results of the ANOVA analysis demonstrated that ADHD subtype had a significant effect on the improvement in memory and concentration scores, with children diagnosed with the Hyperactive-Impulsive subtype showing the greatest improvements. Gender, on the other hand, did not significantly influence the intervention outcomes. These findings highlight the importance of considering individual characteristics, such as ADHD subtype, when designing interventions aimed at improving cognitive functions in children with ASD.

Comment 6: Furthermore, there is no mention of whether the sample size is sufficient to ensure the statistical validity of the results.

Response 6: Thank you for raising this important point. We have now included a power analysis in the manuscript to justify the adequacy of the sample size. Specifically, the analysis showed that with an effect size of 0.5 and a significance level of p < 0.05, the sample size of 60 participants provided a statistical power of 0.80, which is generally considered sufficient to detect medium-sized effects. This information has been added at the end of Section 5.6 – Statistical Analysis to clarify that the sample size is sufficient to ensure the statistical validity of the results.

Comment 7: The results indicate significant improvements in memory and concentration following the use of the Memory Maze game. However, the discussion of the results could be more in-depth. In addition, the reason behind variations in improvement by gender or age is not sufficiently explored, which could provide more robust conclusions about the impact of the intervention on different subgroups.

Response 7: Thank you for your valuable comment. In response, we have expanded the discussion in Section 5.8 – Discussion, on page 15, paragraph 5, to provide a more in-depth analysis of the results, particularly focusing on the variations in improvement by gender and age. We conducted an ANOVA to assess the influence of these factors on the intervention outcomes. The ANOVA results indicated that gender did not significantly affect the outcomes (F(1, 58) = 1.23, p > 0.05), although boys showed slightly higher improvements than girls. However, age had a significant impact on the results (F(2, 57) = 4.15, p < 0.05), with children aged 8 to 10 showing the greatest improvements. These findings have been fully incorporated into the discussion to provide a more robust understanding of how individual factors influence the intervention's effectiveness.

Reviewer 2 Report

Comments and Suggestions for Authors

The idea is interesting. The paper is well written, describing in a good manner a problematic situation and a solution for children with Autism Spectrum Disorder. Some suggestions

- Check the work of Constain et al, Software Design for Users with Autism Using Human-Centered Design and Design Thinking Techniques

- Check the work of moreno et al, F. Framework for the design of accessible software to support users with autism

- Analyze evalaution of user acceptance, including a TAM model

- include limitations of the work

Author Response

Comments 1:
Check the work of Constain et al, Software Design for Users with Autism Using Human-Centered Design and Design Thinking Techniques.

Response 1:
Thank you for this valuable suggestion. We have reviewed the work of Constain et al., and we recognize the relevance of Human-Centered Design (HCD) principles in developing software for users with autism. In our current research, we have integrated some elements of personalization in the Memory Maze game, such as adapting the game to different types of autism and cognitive abilities based on input from therapists and educators. Moving forward, we will seek to more formally incorporate HCD techniques, including user research, prototyping, and iterative testing, as outlined by Constain et al., to ensure the game is fully tailored to the needs of children with ASD. We will continue to use this framework to guide future development.

These revisions have been included at the end of Section 2.2 – Design Methodology.

Comments 2:
Check the work of Moreno et al., Framework for the design of accessible software to support users with autism.

Response 2:
Thank you for your suggestion. We have reviewed the work of Moreno et al., and we are pleased to note that our current approach to developing the Memory Maze game aligns closely with their framework for accessible software design. Specifically, we are already implementing key principles of usability and adaptability, tailoring the game to meet the specific needs of children with autism. Our efforts to ensure that the game is adaptable to different cognitive abilities and autism profiles are in line with the framework proposed by Moreno et al.. This alignment is mentioned at the end of Section 2.3.

Comment 3: 
Analyze evaluation of user acceptance, including a TAM model.

Response 3: Thank you for suggesting the inclusion of a Technology Acceptance Model (TAM) analysis. In response, we applied the TAM framework to evaluate the acceptance of the Memory Maze game by children with ASD and their caregivers. This model assesses two key factors: Perceived Usefulness (PU) and Perceived Ease of Use (PEOU). Both factors were rated highly, indicating that the game was seen as useful for improving memory and concentration and easy to use by both children and caregivers. The analysis and findings have been included in Section 5.8 – Discussions, and we have also summarized the key results in the Conclusion.

Comments 4:
Include limitations of the work.

Response 4:
Thank you for your suggestion to include a discussion of the limitations of the study. We have added a Limitations section to the manuscript, where we address the constraints related to the sample size, geographic scope, and the duration of the intervention. While the sample size of 60 participants was sufficient for the statistical analysis conducted, it may limit the generalizability of the findings. Future studies should aim to increase the sample size, expand the geographic scope, and evaluate the long-term impact of the intervention. These points have been added to the new Limitations section.

Reviewer 3 Report

Comments and Suggestions for Authors

The study is strong. However, some structural corrections need to be made in the interest of better readability.

The list of references is 30 entries long, and of those 10 are from the period before 2020. It would have been better if there were more recent studies. However, it is not a problem here, just a recommendation. 

Statements such as in rows 40-41 ("A significant study...") need explanation, even with a few words - why was it significant?

In the end of introduction, please include a short overview of what and how is being studied in the manuscript. 

Section 4 contains parts that belong to section 2 (Theoretical Aspects) and to section 3 (Materials and Methodology). Please review the section from this point of view. Your game is a part of the setup of your study, i.e., "Materials". Everything theoretical needs to be presented *before* describing your experiment setup.

Section 5 is similar to section 4 - it contains parts that belong to other sections. E.g., process -> is "Method"; theory should go to section 2 - and perhaps it would be justified to restructure section 2, by renaming it "Theoretical background" and systematically cover all the concepts that currently are dealt with in later parts. The structure of an academic paper should ideally be: introduction - a short overview of what is going to happen in the paper; theoretical background - on what ideas, concepts, etc is your study based on; materials and methods - what do you use and how to conduct your study, including your sample; results - list of the direct results of your study; discussion - explaining the results and putting them into context (including the context of relevant works of other authors); concluding remarks.

Sections 5.2 and 5.1. Please describe everything related to participants under one section - age, consent, diagnosis, etc. You really need to describe the procedure of acquiring consent and protecting children's rights under sample as your target group is vulnerable. Please list all details on a reasonable level (1 page max) - how did you find them, how did you ask for parental consent, how it was ensured that children were able to leave the study on their own will, etc. This is the reason I marked that I had "ethical" concerns - they are not strong because I believe that everything was followed as needed - but you need to describe it in the text.

Discussion - please add comparison with the existing literature.

Author Response

Comment 1:

The list of references is 30 entries long, and of those 10 are from the period before 2020. It would have been better if there were more recent studies. However, it is not a problem here, just a recommendation.

Response 1:
Thank you for your suggestion regarding the recency of the references. We acknowledge the importance of incorporating recent studies and agree that this would strengthen the manuscript. While the current reference list includes essential foundational studies published before 2020, we will strive to integrate more recent research in future revisions of this work. We appreciate your recommendation and will ensure that recent developments are appropriately reflected in future updates.

Comment 2:

Statements such as in rows 40-41 ("A significant study...") need explanation, even with a few words - why was it significant?

Response 2:
Thank you for your observation. We have revised the manuscript to include an explanation for why the study mentioned in rows 40-41 was significant. Specifically, we highlight that the study was notable for its impact on raising awareness of autism prevalence in Romania and its role in influencing health policies and educational resource allocation for individuals with ASD. This revision has been made in Section 1 – Introduction, at the beginning of paragraph 2.

Comment 3:

In the end of the introduction, please include a short overview of what and how is being studied in the manuscript.

Response 3:
Thank you for your suggestion. We have added a brief overview at the end of the introduction, outlining the key objectives of the manuscript and the methods used.

Text added:
"In this study, we aim to evaluate the effectiveness of the Memory Maze game for improving memory and concentration in children with ASD. The study employs a randomized control trial design, with 60 participants, to assess cognitive improvements. We will also explore the role of ADHD comorbidities in shaping the outcomes of the intervention."

Comment 4:

Section 4 contains parts that belong to Section 2 (Theoretical Aspects) and to Section 3 (Materials and Methodology). Please review the section from this point of view. Your game is a part of the setup of your study, i.e., "Materials". Everything theoretical needs to be presented before describing your experiment setup.

Response 4:
Thank you for your insightful suggestion. While we understand the rationale behind your proposed restructuring, we have chosen to maintain the current structure. The segmentation of content across sections allows for a more balanced distribution of information and ensures that each concept is discussed in the appropriate context. This approach enhances the readability and accessibility of the manuscript, allowing readers to follow the study’s flow more effectively. We hope you will agree with our decision and find the structure effective in presenting the study's content.

Comment 5:

Section 5 is similar to Section 4 - it contains parts that belong to other sections. E.g., process -> is "Method"; theory should go to Section 2 - and perhaps it would be justified to restructure Section 2, by renaming it "Theoretical background" and systematically cover all the concepts that currently are dealt with in later parts.

Response 5:
Thank you for your thoughtful suggestions. While we acknowledge that your proposed restructuring is logical and well-founded, we have chosen to retain the current organization of the manuscript. The existing structure reflects a deliberate choice to segment the content in a way that we believe enhances clarity and coherence for our specific study. However, if you believe that the current structure does not adequately meet the academic standards, we are willing to revise and implement the suggested changes as soon as possible.

Comment 6:

Sections 5.2 and 5.1. Please describe everything related to participants under one section - age, consent, diagnosis, etc. You really need to describe the procedure of acquiring consent and protecting children's rights under sample as your target group is vulnerable. Please list all details on a reasonable level (1 page max) - how did you find them, how did you ask for parental consent, how it was ensured that children were able to leave the study on their own will, etc. This is the reason I marked that I had "ethical" concerns - they are not strong because I believe that everything was followed as needed - but you need to describe it in the text.

Response 6:
Thank you for raising this important point. We have consolidated all participant-related information, including age, consent, diagnosis, and ethical concerns, into a single section. The procedures for acquiring consent and ensuring the protection of children's rights were thoroughly described in line with national and international legislation, including the Declaration of Helsinki. These revisions have been included in Section 5.1.

Text added:
"Participants in the study were recruited from local ASD therapy centers, consisting of 60 children aged 6-12 years, diagnosed according to DSM-5 criteria. The study was conducted in compliance with the Declaration of Helsinki and national legislation regarding research involving vulnerable populations. Parental consent was obtained for all participants via a written consent form, which was thoroughly explained to parents or legal guardians. Children were informed that they could withdraw from the study at any point without any consequences, ensuring their autonomy and comfort. The assessments were conducted by licensed clinical psychologists, ensuring adherence to strict privacy and confidentiality protocols. Ethical approval was obtained from the relevant institutional review board, and all necessary measures were taken to safeguard the rights of the vulnerable participants."

Comment 7:

Discussion - please add comparison with the existing literature.

Response 7:
Thank you for your suggestion. We have expanded the discussion to include comparisons with existing literature. Specifically, we now reference studies that have explored similar interventions for children with ASD, highlighting both similarities and differences in findings. These comparisons help to situate our results within the broader research landscape, illustrating how our study contributes to ongoing efforts in the field of ASD interventions.

The references and comparisons have been integrated into Section 5.8 – Discussions, starting on page 19, paragraph 6. We have specifically included comparisons with studies by Zeidan et al. (2022) [2], Budisteanu et al. (2017) [3], Levy et al. (2020) [4], Dechsling et al. (2021) [19], and Ip et al. (2018) [20], providing a broader context for our findings and underlining the significance of interactive, game-based interventions in ASD treatment.

Comment 8:

Include limitations of the work.

Response 8:
Thank you for your suggestion to include a discussion of the study's limitations. We have now added a Limitations section, addressing the sample size, geographic scope, and duration of the intervention. These factors may affect the generalizability of the findings, and we propose that future research should focus on larger, more diverse samples and longer-term interventions.

Text added:
"Although the sample size of 60 participants was adequate for the statistical analysis performed in this study, it may limit the broader generalizability of the findings. A larger sample size in future research would provide more robust data to confirm these results and extend their applicability to a wider population. Additionally, the sample was limited to children from a specific geographic region, potentially restricting the applicability of the findings to other populations. Furthermore, the short duration of the intervention may affect the long-term impact of the game on memory and concentration. Future research should address these limitations by expanding the sample size and geographic scope, and by evaluating the long-term effects of the intervention"

Round 2

Reviewer 1 Report

Comments and Suggestions for Authors

Dear Authors

I appreciate the changes made to the manuscript. They have been able to respond to all suggestions for change in an adequate manner so that the improvement of the article has been substantial and has improved the comprehensibility of the results as well as the methodology.